# The Brown Alga *Bifurcaria bifurcata* Presents an Anthelmintic Activity on All Developmental Stages of the Parasitic Nematode *Heligmosomoides polygyrus bakeri*

**DOI:** 10.3390/pathogens12040540

**Published:** 2023-03-30

**Authors:** Morgane Miclon, Élise Courtot, Fabrice Guégnard, Océane Lenhof, Leslie Boudesocque-Delaye, Maria Matard-Mann, Pi Nyvall Collén, Philippe Castagnone-Sereno, Cédric Neveu

**Affiliations:** 1Institut National de Recherche pour l’Agriculture, l’Alimentation et l’Environnement (INRAE), Université de Tours, ISP, 37380 Nouzilly, France; 2EA 7502 Synthèse et Isolement de Molécules BioActives (SIMBA), Faculté de Pharmacie, Université de Tours, 31 Avenue Monge, 37200 Tours, France; 3Olmix S.A., 56580 Bréhan, France; 4Institut National de Recherche pour l’Agriculture, l’Alimentation et l’Environnement (INRAE), Université Côte d’Azur, CNRS, ISA, 06903 Sophia Antipolis, France

**Keywords:** anthelmintic, natural compound, algae, gastrointestinal parasite, nematode, *Heligmosomoides polygyrus bakeri*, Bifurcaria bifurcata

## Abstract

The current control of gastrointestinal (GI) parasitic nematodes mainly relies on the widespread use of anthelmintics, which has inevitably led to resistance. Therefore, there is an urgent need to find new sources of antiparasitic compounds. Macroalgae represent a rich source of active molecules and are widely described as having medicinal properties. In the present study, we investigated the potential anthelmintic activity of aqueous extracts from three species of algae (*Bifurcaria bifurcata, Grateloupia turuturu* and *Osmundea pinnatifida*) on the murine parasite *Heligmosomoides polygyrus bakeri*. Using a set of complementary in vitro tests, including larval development assays, egg hatching tests and nematicidal activity assays on larvae and adults, we report the nematicidal activity of aqueous extracts of *B. bifurcata*. In addition, aqueous extract fractionation using liquid/liquid partitioning with a solvent of increasing polarity was performed in order to identify the groups of active molecules underlying the anthelmintic activity. Non-polar extracts (heptane, ethyl acetate) demonstrated high anthelmintic potential, highlighting the role of non-polar metabolites such as terpenes. Here, we highlight the strong anthelmintic potential of the brown alga *B. bifurcata* on a mouse model of GI parasites, thus confirming the strong interest in algae as natural alternatives for the control of parasitic nematodes.

## 1. Introduction

Gastrointestinal nematodes are a major concern for both human and animal health worldwide. In livestock, these parasites affect the health of animals by causing inappetence, diarrhea, anemia and, in severe cases, death. Parasitism also impairs productivity and causes major economic losses in livestock production [1,2]. In the absence of effective alternative strategies, the control of parasitic nematodes affecting human or animal health is essentially based on the use of chemical anthelmintics. However, their intensive and sometimes inappropriate use has led to the development of anthelmintic resistance [3]. Therefore, there is an urgent need to find new and complementary solutions for the sustainable control of helminths.

In order to identify new potential anthelmintic compounds, an increasing number of investigations are now focusing on molecules of natural origin for the control of GI nematode populations (for a review, see [4]). Recent studies have shown the strong potential of algae through their anti-inflammatory, antimicrobial, anti-cancer and anti-parasitic activities [5,6,7,8]. One of the most striking examples of the efficacy of seaweeds as potent anthelmintics is the use of the red alga *Digenea simplex* for the treatment of ascariasis in Japan for many centuries [9].

However, due to the limited geographical distribution of *Diginea simplex* [10] and the absence of an efficient cultivation system, the identification and selection of alternative algae species exhibiting similar anthelmintic properties are now urgently needed.

In the present study, we decided to focus our attention on the putative anthelmintic activity of the algae *Bifurcaria bifurcata, Grateloupia turuturu* and *Osmundea pinnatifida.* These algae have been chosen based on different criteria: (i) biological activity already described in the literature, (ii) reduced toxicity on mammalian cells and (iii) amenability for cultivation or wide availability ensuring a potential industrialization. The algal species *G. turuturu* is a red alga endemic to Japan, where it is traditionally used in food [11]. It has been introduced to both sides of the Atlantic Ocean, but now has a global distribution and has even become invasive [12].This alga seems to produce many active compounds due to its invasive capacity. In particular, it has been shown to present antibacterial [13] and anti-microfouling activity [12]. The *O. pinnatifida* alga is also a red alga that has a worldwide distribution, but it is mainly present on the European coasts. It is widely used in cooking due to its very powerful aroma (common name: pepper dulse). It has several biological activities, including antifungal activity [14]. This alga is already commercially available, and encouraging results concerning its potential cultivation have been reported [15]. Finally, the algal species *B. bifurcata* is a brown alga widespread on the European Atlantic coast, distributed from Morocco to Ireland [16]. It has been studied for several biological activities [17,18,19], including anti-protozoal properties, for which several active molecules have been identified [20,21].

In order to investigate the putative anthelmintic activity of these three algae species, we used the murine parasite species *Heligmosomoides polygyrus bakeri* as a model. This parasite species is closely related to other trichostrongylid parasitic nematodes of veterinary importance, such as *Haemonchus contortus* or *Teladorsagia circumcicta* [22]. It is therefore routinely used for the screening of anthelmintic molecules in the academic sector [23,24].

In the present work, we used a panel of in vitro assays to investigate the antiparasitic activity of *B. bifurcata, G. turuturu* and *O. pinnatifida* on different life stages of *H. polygyrus bakeri.* We confirmed the potent anthelmintic activity of aqueous extracts of *B. bifurcata* on different life stages of the parasite and identified the organic fraction underlying the biological activities.

## 2. Materials and Methods

### 2.1. Algae Collection and Aqueous Extract Preparation

The two red algae *G. turuturu* and *O. pinnatifida* and the brown alga *B. bifurcata* (FR-1) were collected in Roscoff (Bretagne, France, 48°43′53.6″ N 3°59′15.5″ W) in 2019. Algae were washed with sea water and stored for 24–48 h at 4 °C before treatment. Cleaned algae were desalted in osmosis water and stored at −20 °C. To carry out the aqueous extractions, the algae were thawed at room temperature and ground using an ULTRA-TURRAX^®^. The ground algae were incubated for 1 h at RT and then centrifuged for 20 min at 10,000× *g* to collect the supernatant (extract 1). The residual pellet was resuspended in a volume of water equivalent to the weight of the pellet and incubated for one hour at 60 °C in a water bath. After centrifugation at 10,000× *g* for 20 min, the supernatant was recovered (extract 2). The two extracts obtained were lyophilized separately for optimal conservation.

In order to investigate potential modulation of the anthelmintic activity related to the geographical origin and process, batches of *B. bifurcata* were also collected in Peniche (Leiria, Portugal, 39°19′33″ N 9°21′35.4″ W) in 2020. QuintaQuanta/Mermaid Fragrance was the biomass supplier. The seaweeds from Portugal were sent dried. They were not washed before drying, so desalting was carried out during the rehydration phase with osmosis water. Extraction was then carried out in the same way as for frozen seaweed.

All the extracts were resuspended at a concentration of 15 mg/mL in mineral water (Volvic, France), aliquoted and stored at −20 °C until use.

### 2.2. Preparation of Organic Extracts

Freeze-dried algal material (3 g) was solubilized in 30 mL of MilliQ water before extraction and then sequentially extracted three times with each solvent (*n*-heptane, ethyl acetate (EtOAc) and finally *n*-butanol (BuOH): (1:1, *v*/*v*)). All solvents were analytical grade and were purchased from Carlo Erba France. These three different organic extracts as well as the residual water were filtered through filter paper and evaporated to dryness in a rotary vacuum evaporator (Hei-VAP Core, Heidolph Instruments, Schwabach, Germany) at 33 °C, leading to an average recovery of 6 mg (Heptane), 34 mg (EtOAc), 147 mg (BuOH) and 1.7 g (Water) of resulting dried extracts. The dried extracts were dissolved in DMSO (Sigma-Aldrich, Merck, Darmstadt, Germany) at 100 mg/mL for stock solution stored at −20 °C and further diluted as per dose requirement.

### 2.3. Parasite Material

*H. polygyrus bakeri* eggs, larval stages and adult worms were obtained from experimentally infected mice (C57BL/6JRj) under controlled conditions. Experimental infestations, mice maintenance and euthanasia were conducted in strict accordance with the national ethics guidelines and approved by the local ethics committee for animal experimentation (Comité d’Ethique en Expérimentation Animale Val de Loire, CEA VdL N°19) under protocol number APAFIS#21504. Briefly, mice were inoculated orally with 200 L3s of *H. polygyrus bakeri*. Ten days after infection, fecal samples were collected for egg extraction. The collected feces were pooled, mixed with water in a mortar and filtered through different filters measuring 100 µm and 35 µm. The eggs were collected in the 35 µm filter and mixed with kaolin to pellet them. After 5 min centrifugation at 600× *g*, the pellet was resuspended in a solution of NaCl at 360 mg/mL. A second centrifugation of 5 min at 900× *g* was carried out, the supernatant was discarded and the eggs were rinsed with water before being collected and counted.

For collecting adult worms, mice were sacrificed by cervical dislocation and the small intestine was recovered. It was opened along its entire length with a pair of dissecting scissors. The intestine was placed in a filter on top of a glass funnel connected to a harvest tube. Worms migrated through pre-warmed (37 °C) DMEM (Gibco, Thermo Fisher Scientific, Waltham, MA, USA) culture medium from the filter to the harvest tube, as previously described [25].

### 2.4. Hatching Test and Mortality after Hatching Test

One hundred eggs (60 µL) were placed in glass tubes in the presence of the algal extract (30 µL). The tubes were placed at 21 °C for 48 h to induce egg hatching. The test consisted in counting the number of eggs that had not hatched and the number of larvae released after egg hatching. The egg hatching ratio was calculated (larvae/(larvae + eggs)) and results were corrected according to the negative control to obtain the percentage of egg hatching inhibition, as follows:$$Egg\;hatch\;inginhibition=100-\left(\frac{hatching\;ratio\;of\;eggs\;exposed\;to\;extract}{hatching\;ratio\;of\;eggs\;of\;negative\;control}\times 100\right)$$

Among the hatched larvae, the number of live and dead larvae was also counted to obtain the percentage of mortality of post-hatch larvae, which was calculated as follows:$$Percentage\;of\;mortality\;of\;larvae\;after\;hatching=\left(\frac{dead\;larvae}{dead\;larvae+live\;larvae}\right)\times 100$$

Thiabendazol (Sigma-Aldrich, Merck, Darmstadt, Germany) was used as a reference drug for this test at a concentration of 0.5 µg/mL.

### 2.5. Larval Development Assay (LDA)

LDA was performed by taking 7.8 mL of egg suspension, which contained ~1666 eggs/mL, and adding 2 mL of an *E. coli* suspension (150 µg/mL) and 200 µL of Amphotericin B (500 µg/mL) (Sigma-Aldrich, Merck, Darmstadt, Germany). The resulting mix was then dispensed into glass tubes (80 µL per tube). The tubes were placed at 21 °C for 48 h to allow the eggs to hatch properly. At 48 h, 20 µL of Earl’s medium (Sigma-Aldrich, Merck, Darmstadt, Germany) was added, as well as the algae extracts to be tested (50 µL). The tubes were placed back in the incubator at 21 °C for 7 days to allow the proper development of the larvae. The test then consisted in counting the number of young L1/L2 larvae compared to the number of fully developed larvae at L3 stage. The ratio of larval development was calculated (L3/(L3 + L1/L2)) and results were corrected according to the negative control to obtain the percentage of inhibition of larval development, as follows:$$Larval\;development\;inhibition=100-\left(\frac{ratio\;of\;development\;of\;larvae\;exposed\;to\;extract}{mean\;ratio\;of\;development\;of\;larvae\;of\;negative\;control}\times 100\right)$$

Ivermectin (Sigma) was used as a reference drug for this test at a concentration of 1 µM.

### 2.6. Survival Test

Adult worms were placed in 48-well plates containing 120 µL of DMEM medium (Gibco, Thermo Fisher Scientific, Waltham, MA, USA) supplemented with antibiotics (Penicillin 100 U/mL and Streptomycin 100 µg/mL from Gibco) and 60 µL of algal extracts to obtain the different final concentrations (5 mg/mL, 1 mg/mL, 0.5 mg/mL and 0.1 mg/mL). The worms were placed at 37 °C and the number of survivors was counted every day for six days. Worms were considered dead when they did not react to contact stimulation (being poked with a worm pick).

### 2.7. Statistical Analyses

R Studio 4.2.1 was used for statistical analysis of mean, median, calculation of IC50 by nonlinear regression and survival analysis by using the Kaplan–Meier method. Due to the distribution of the data, which did not follow a normal distribution, and the non-homogeneity of the variances, Kruskal–Wallis test and Multiple Comparison Test (Dunn’s test and Wilcoxon test) were used.

## 3. Results

### 3.1. Larval Development Assay (LDA) Reveals Anthelmintic Activity of Aqueous Algal Extracts on H. polygyrus bakeri

The potential larval development inhibition of *H. polygyrus bakeri* larvae by aqueous extracts of *B. bifurcata, G. turuturu* and *O. pinnatifida* was investigated using water as a negative control and ivermectin 1 µM (a reference anthelmintic drug) as a positive control.

In water, the development from the L1/L2 to the L3 stage was approximately 97%, whereas the 1 µM ivermectin application completely inhibited larval development. As an additional control, we also investigated the potential effect of salt (NaCl) to discriminate between the putative activity of bioactive compounds and the salt content of seaweeds. At 5 mg/mL, the NaCl solution had no effect on larval development (similar to the control in water only).

For each algae species, in order to take into account the potential effect of the extraction temperature on the biological activity, two aqueous extracts (cold and hot extractions) were tested (Figure 1A). None of the 5 mg/mL aqueous extracts of *G. turuturu* and *O. pinnatifida* had an impact on the development up to the L3 stage. In sharp contrast, both 5 mg/mL extracts from *B. bifurcata* induced a complete inhibition of the larval development. Their respective IC50 values were 0.68 ± 0.02 mg/mL for “cold” extract 1 and 0.97 ± 0.03 mg/mL for “hot” extract 2, and they were significantly different (*p* < 0.001) (Figure 1B). Importantly, the anthelmintic activity of *B. bifurcata* at 5 mg/mL was further confirmed on algae batches collected in 2019 and 2021 that were frozen either directly or after 24 h at +4 °C or 48 h at +4 °C (Appendix A). Overall, these results demonstrated that, among the three algal species screened with the LDA, only *B. bifurcata* presented a strong, reproducible and dose-dependent anthelmintic activity on larval stages of *H. polygyrus*. Since both the “cold” and “hot” extracts presented significant anthelmintic activity, we decided to focus our attention on *B. bifurcata* cold extracts, which exhibited the lowest IC50.

### 3.2. B. bifurcata Inhibits Egg Hatching and Has Nematicidal Activity on H. polygyrus bakeri Adult Worms

In order to further investigate the anthelmintic activity of *B. bifurcata* on other life stages of the parasite, complementary tests were performed on eggs and adult worms.

First, egg hatching tests (48 h) were carried out using water as the negative control and thiabendazole (a reference anthelmintic drug for egg hatch tests [26]) as the positive control (Figure 2A). The application of *B. bifurcata* aqueous extract (5 mg/mL) induced a 28% inhibition of egg hatching in comparison with the negative control (*p* < 0.001).

Strikingly, strong nematicidal activity was also observed on freshly hatched larvae, with 67% (*p* < 0.001) of mortality (Figure 2B). 

Second, the direct nematicidal activity of *B. bifurcata* was also investigated on adult worms via a survival test. The test lasted 6 days, during which the survival rate of the adult worms was 100% in the culture medium. The alga *B. bifurcata* aqueous extract was applied at four different concentrations (from 5 mg/mL to 0.1 mg/mL; Figure 3). For the highest concentration, all worms died within 24 h. For the 1 mg/mL concentration, we observed a low survival rate of 10% after three days. Finally, nematicidal activity was still detectable at a concentration of 0.5 mg/mL, with a survival rate of 30% after 6 days. These results strongly support the direct nematicidal activity of *B. bifurcata* on adult *H. polygyrus bakeri* worms. It is worth noting that the test was not performed over a longer period of time because the survival rate of the worms in control conditions started to decrease.

Taken together, these results clearly demonstrate that the aqueous extracts of *B. bifurcata* algae exhibit anthelmintic activity against the murine parasite *H. polygyrus bakeri.*

### 3.3. The Anthelmintic Activity of B. bifurcata Could Vary Depending on the Extraction Process and/or Geographical Origin

Working with algal biomass, it was important to investigate the reproducibility of the results as a function of geographical origin and extraction process. Indeed, these two factors are well known to influence the phytochemical composition of extracts and thereby their biological activity. Therefore, we used LDA assays to compare the anthelmintic activity of *B. bifurcata* from France (frozen before aqueous extraction) with the *B. bifurcata* batches harvested on the Portuguese coast (dried before aqueous extraction). In order to monitor the potential impact of the combined geographical and process factors, dose-response curves were generated for both types of extracts (Figure 4). Our results confirm dose-dependent anthelmintic activity in both *B. bifurcata* batches and highlight variability in efficacy (EC50 = 0.68 ± 0.02 mg/mL for the French frozen batch vs. EC50 = 1.52 ± 0.10 mg/mL for the Portuguese dried batch, *p* < 0.001) which could reflect either the geographical origin of the algae and/or the impact of the drying process prior to aqueous extraction.

In order to further explore the phytochemical family potentially contributing to the variability of the anthelmintic activity of *B. bifurcata*, we carried out organic extractions on the aqueous extracts obtained from algae batches from France (frozen) and Portugal (dried) at 1 mg/mL. At this concentration, the aqueous extract from France was still highly active, whereas its counterpart from Portugal was almost inefficient, thus increasing our ability to detect the corresponding chemical component responsible for the differential anthelmintic activity by subsequent fractionations. The fractionation of the aqueous extracts was performed sequentially with three distinct solvents (heptane, EtOAc and BuOH), and their respective anthelmintic properties were monitored using LDA as previously described (Figure 5). The organic extracts (heptane, EtOAc and BuOH) and the residual aqueous fraction were tested at 1 mg/mL. For both the French and Portuguese algae, the heptane and ethyl acetate extracts induced a total inhibition of larval development. Strikingly, whereas the butanolic extract from the French algae presented a high efficacy on larval development (99%), its Portuguese counterpart only induced a 13% inhibition, thus laying the basis for a potential explanation for their differential activity. Importantly, the residual aqueous fractions did not show any remaining activity, indicating that the active molecules were completely recovered by the organic solvents.

The anthelmintic activity of the organic fractions was further investigated using a post-hatching larval mortality test (Table 1). The organic extracts were tested at different concentrations. Whereas the BuOH extracts and the residual aqueous fractions showed a weak impact on the mortality of the larvae even at 3 mg/mL, the ethyl acetate extracts were efficient at 1 mg/mL, and the heptane extracts had full activity at 0.5 mg/mL. These results indicate that the active molecules are mainly present not only in the heptane extract, but also in the ethyl acetate extract. Interestingly, there were no significant differences observed between the activities of the organic extracts of the French and Portuguese algae. In contrast to the LDA results, there was only weak activity in the BuOH fraction of French algae regarding larval mortality. These results suggested that the molecules involved in the inhibition of larval development could differ from those involved in larval mortality.

## 4. Discussion

In the present study, we investigated the putative anthelmintic properties of three algae species based on their wide availability, low toxicity and possible large-scale production. Among the three species tested on the murine parasite *H. polygyrus bakeri*, *B. bifurcata* was the most potent to inhibit the larval development with 100% efficacy at a concentration of 5 mg/mL. This alga also showed effects on egg hatching (28% inhibition at 5 mg/mL) and direct nematicidal activity on larvae (67% efficacy at 5 mg/mL) and adult worms (70% efficacy at 0.5 mg/mL in 6 days).

Over the past twenty years, the number of studies dealing with the anthelmintic activity of natural products has increased considerably [27]. The anthelmintic potential of aqueous macroalgal extracts has been tested on the plant-parasitic nematode *Meloidogyne incognita* [28]. While the results showed the activity of distinct macroalgal species on the inhibition of egg hatching and the mortality of juvenile larvae, it should be noted that the efficient concentrations reported in the study were well above 5 mg/mL. However, our work concerning the screening of the anthelmintic activity of aqueous macroalgae extract on mammalian parasitic nematodes is, to our knowledge, quite unique. Indeed, the large majority of previous studies evaluating the antiparasitic potential of plants or algae on gastrointestinal nematodes mainly focused on organic extracts [27]. It is nevertheless tempting to compare the results we obtained with *B. bifurcata* to the anthelmintic efficacy previously reported from plant aqueous extracts. In the present study, we observed 100% mortality of adults of *H. polygyrus bakeri* in 24 h with an aqueous extract of *B. bifurcata* at 5 mg/mL, whereas, for example, on *Haemonchus contortus*, Eguale et al. noticed a mortality rate of only 29% in 24 h with an aqueous plant extract at 8 mg/mL, and Lone et al. used much higher concentrations (25 mg/mL and 50 mg/mL) to observe an effect on adult mortality in less than 24 h [29,30]. However, such a comparison should be taken with care, as we cannot rule out that, despite their close phylogenetic relationship, *H. polygyrus bakeri* and *H. contortus* could also present a differential sensitivity to natural compounds.

In the present work, we demonstrated that heated extracts still presented biological activity, thus demonstrating that active compounds are not affected by temperatures up to 60 °C. Because it has been previously reported that temperature can impact the stability of some molecules such as algae polyphenols [31], we could speculate that the active molecules either are present in large quantities or may not be affected by temperature, thus highlighting the important stability of the soluble anthelmintic compounds of *B. bifurcata*.

Interestingly, we highlighted a difference in anthelmintic activity between *B. bifurcata* from France (frozen) and Portugal (dried). The aqueous fractions from the algae batches from Portugal had lower activity in comparison with their counterparts from France. However, the Portuguese batches kept about 30% of the inhibition of larval development, which tends to indicate that the active molecules were still present but in lower quantities. This hypothesis was also confirmed by the comparison of the mass balances of the fractionation of the two batches of algae. Indeed, the BuOH fraction represented 3.1% of the global mass for the French algae, whereas it represented only 2.6% for the Portuguese ones (Appendix A). In order to further investigate this hypothesis, we performed different organic extractions (heptane, ethyl acetate and butanol) and characterized their respective biological activities. Our results showed that the fractions carrying the main biological activity were those obtained with heptane and EtOAc, independently from their origin and the process used before the aqueous extractions (i.e., frozen or dried). This confirmed the presence of active molecules in the Portuguese algae batch. It is noteworthy to mention that the active molecules harboring an antiparasitic activity against the protozoans *Trypanosoma* and *Plasmodium* previously identified in *B. bifurcata* were also extracted using EtOAc [21]. Interestingly, our results are also in agreement with a previous study performed on brown algae *Saccharina latissima* and *Laminaria digitata*, showing that the organic fraction extracted with another alkane solvent (i.e., hexane) also presented larvicidal activity against two species of GI parasites (sheep (*Teladorsagia circumcincta*) and swine (*Ascaris suum*)) [8]. Clearly, our results pave the way for comparative analyses aiming to determine if similar compounds could be involved in the anthelmintic activity of different brown algae species. Importantly, the results of the different organic fractions on the larval development test also highlighted an interesting feature concerning the butanol extract. Indeed, we observed a major difference in activity between the BuOH fraction of the French and Portuguese algae (99% and 13% of larval development inhibition, respectively), suggesting that the anthelmintic properties of *B. bifurcata* could rely on different molecule families. This confirms the particular interest in using *B. bifurcata*’s aqueous extracts as potential anthelmintic sources, as they combine a range of active compounds leading to putatively distinct modes of action. Bonde et al. showed that the anthelmintic activity of individual algal compounds is moderate in comparison with the activity of the fractions from which they have been obtained [8]. In that respect, it is tempting to speculate about a putative synergic effect between the different active compounds present in both the heptane and BuOH extracts of *B. bifurcata*.

The cytotoxicity of *B. bifurcata* extracts has been investigated in previous studies, highlighting some variable cytotoxic effects depending on the active molecules [32]. In that respect, a detailed analysis of the cytotoxicity of our different extracts will be performed as a preliminary step before in vivo studies.

## 5. Conclusions

Taken together, our results demonstrate that the aqueous extracts of the alga *B. bifurcata* contain bioactive compounds with in vitro anthelmintic activity on the different life stages of the murine parasite *H. polygyrus bakeri*. This study opens the way for both bio-guided fractionation assays and mechanistic approaches to decipher the molecular mode of action underlying the anthelmintic properties of the algae extracts.

## Figures and Tables

**Figure 1 pathogens-12-00540-f001:**
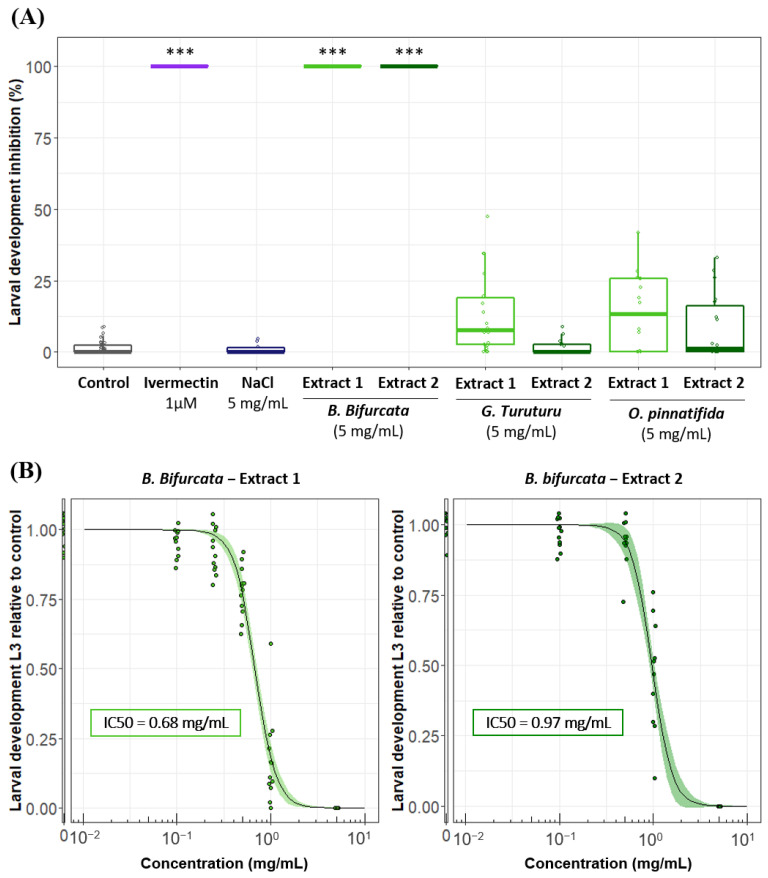
Inhibition of development of *H. polygyrus bakeri* larvae by aqueous extracts of different macroalgae. (**A**) Percentage inhibition of development of *H. polygyrus bakeri,* from L1–L2 to L3, in presence of water (control), ivermectin (1 µM), NaCl (5 mg/mL) and different extracts from *Bifurcaria bifurcata, Grateloupia turuturu* and *Osmundea pinnatifida* (cold extraction (1) and hot extraction (2)) at 5 mg/mL. Results are from at least three independent experiments, with *n* = 6 per sample each time. Statistical analyses were performed using a Kruskal–Wallis test and Multiple Comparison Test with *** *p* < 0.001 compared to the control. (**B**) Dose-response curves for extracts 1 and 2 from *B. bifurcata* for the inhibition of L3 development. Solid line stands for the fitted log-logistic regression curve, and shaded area indicates 95% confidence interval. Results are from two independent experiments, with *n* = 6 per sample each time.

**Figure 2 pathogens-12-00540-f002:**
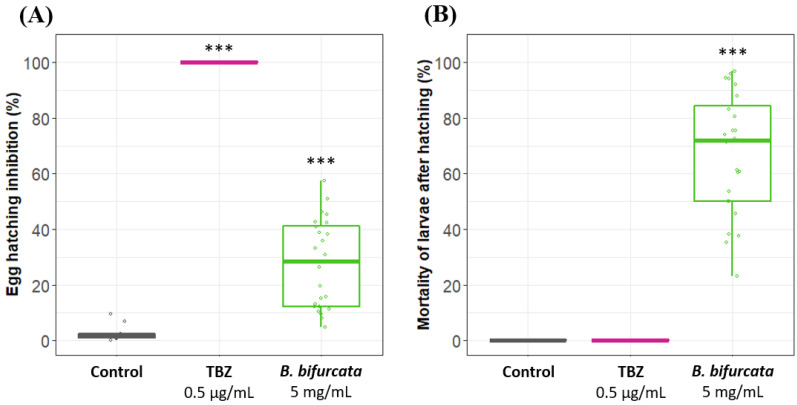
Activity of an aqueous extract of *Bifurcaria bifurcata* on eggs and the first larval stages of *H. polygyrus bakeri*. Effect of water (control), thiabendazole (TBZ) at 0.5 µg/mL and *B. bifurcata* extract at 5 mg/mL on *H. polygyrus bakeri.* (**A**) Egg hatching inhibition after 48 h and (**B**) mortality rate of larvae after hatching. Statistical analyses were performed using a Kruskal–Wallis test and Multiple Comparison Test with *** *p* < 0.001. Results are from three independent experiments, with at least *n* = 6 per sample each time.

**Figure 3 pathogens-12-00540-f003:**
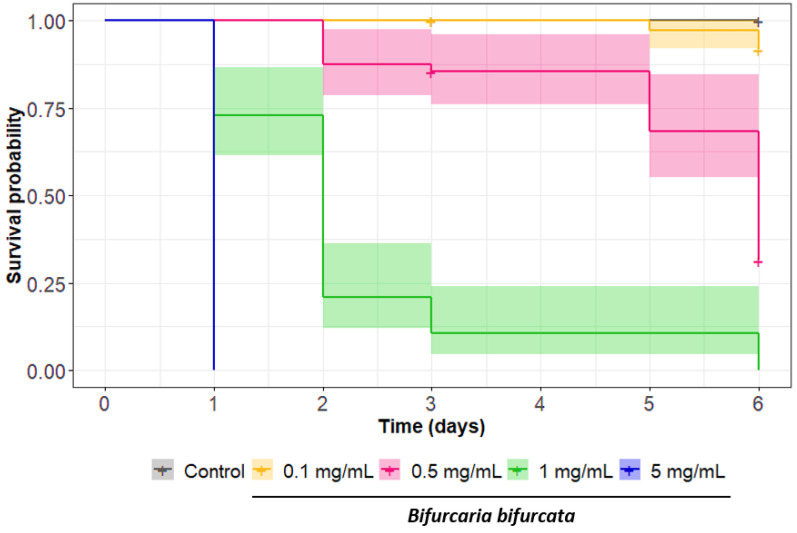
Survival curves for *H. polygyrus bakeri* adults exposed to different concentrations of aqueous extract of *Bifurcaria bifurcata* for 6 days. Survival was plotted using the Kaplan–Meier method, and shaded area indicates 95% confidence interval. Results are from four independent experiments, with at least *n* = 12 per sample each time.

**Figure 4 pathogens-12-00540-f004:**
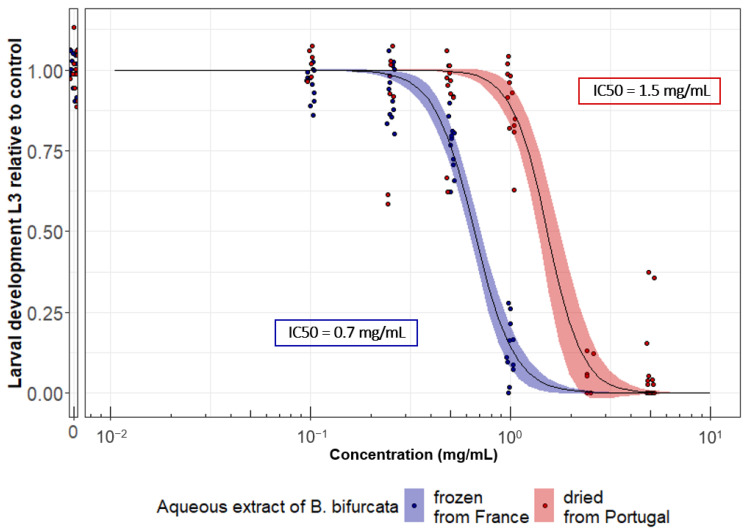
Comparison of different batches of *B. bifurcata* for the inhibition of the larval development of *H. polygyrus bakeri.* Dose-response curves of development of *H. bakeri,* from L1–L2 to L3, with different batches of *B. bifurcata*. The two batches come from algae collected in France and frozen before extraction and from algae collected in Portugal and dried before extraction. Solid line stands for the fitted log-logistic regression curve, and shaded area indicates 95% confidence interval. Results are from two independent experiments, with at least *n* = 6 per sample each time.

**Figure 5 pathogens-12-00540-f005:**
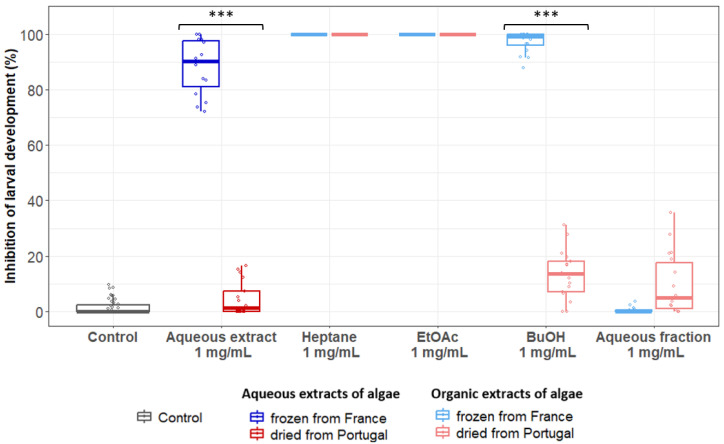
Comparison of the efficacy of different extracts of *B. bifurcata* from France (frozen) and Portugal (dried) in inhibiting the development of *H. polygyrus bakeri* larvae up to the L3 stage. The aqueous extracts, the different organic extracts (heptane, ethyl acetate and butanol) and the residual aqueous fraction were tested at 1 mg/mL. Results are from at least three independent experiments, with *n* = 6 per sample each time. Statistical analyses were performed using a Kruskal–Wallis test and Multiple Comparison Test with *** *p* < 0.001.

**Table 1 pathogens-12-00540-t001:** Comparison of the nematicidal activity of different extracts of *B. bifurcata* from France (frozen) and Portugal (dried) on freshly hatched larvae of *H. polygyrus bakeri*.

Condition	Extract	Concentration (mg/mL)	Geographical Location/Process	Mortality (% ± SD)	*p*-Value France (Frozen) vs. Portugal (Dried)
Control	-	0	-	0.065 ± 0.34	-
*Bifurcaria bifurcata*	Aqueous extract	3	France (frozen)	12 ± 7.15	3.1 × 10^−5^
			Portugal (dried)	0.318 ± 0.74	
	Heptane	0.5	France (frozen)	100 ± 0.00	1
			Portugal (dried)	99.8 ± 0.51	
	EtOAc	1	France (frozen)	73.2 ± 17.90	1
			Portugal (dried)	55.3 ± 31.20	
	BuOH	3	France (frozen)	2.49 ± 3.42	1
			Portugal (dried)	3.54 ± 4.07	
	Residual aqueous fraction	3	France (frozen)	0 ± 0.00	nd
			Portugal (dried)	0 ± 0.00	

Results are from three independent experiments, with *n* = 6 per sample each time. Values are the mean of the mortality (%) ± Standard Deviation; nd: not determined.

## Data Availability

Not applicable.

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
