# Peer review of "The Brown Alga Bifurcaria bifurcata Presents an Anthelmintic Activity on All Developmental Stages of the Parasitic Nematode Heligmosomoides polygyrus bakeri"

_pathogens, 2023, doi:10.3390/pathogens12040540_

Round 1
Reviewer 1 Report
This interesting study is generally well-done and well-written, with only minor attention needed to make sure verb tenses match up (I congratulate the authors for writing well in a second language). One minor comment in this regard: always put a space between a number and a unit (e.g., 4 ml).
I have one rather significant recommendation. The authors note that the composition of NP extracts from plants may vary with location, time of harvest and processing protocol, and indeed show that this is true for the current situation. Although they note that this particular alga has previously been shown to not be cytotoxic in mammalian cell cultures, in light of the above background, it would have been advisable to confirm this with the current extracts.
A few minor points:
1. Provide concentration for amphotericin B and source for all chemicals; note that drug names should not be capitalized.
2. Some experimental details are missing; the legend for Figure 1 includes a statement about the number of independent biological replicates and the number of replicates within a single experiment, but this information is missing for the other experiments and should be added.
3. More statistical details are needed in some places; line 183 compares figure of 0.68 and 0.97 g/ml without an explicit estimate of variance in these means (even the numerical 95% confidence limits around the means) or whether they are statistically significantly different. More information about this needs to be added.
I hope the authors are able to follow up this work with in vivo experiments and chemical identification of the active component(s) of the extracts, which will greatly increase the value of this project (I am sure that is in the plans).
Author Response
"Please see the attachment."

Reviewer 2 Report
This is a well written manuscript evaluating the aqueous extracts of three algae species in vitro, with Bifurcaria bifurcate showing the highest activity. However, I have a few comments summarized below
Introduction
Line 53: I would add here that Bifurcaria bifurcate has shown antiprotozoal activity
Line 73: rather academic sector than industry
Line 75 rather used than developed, as these are standard assays
Methods
2.6. for how long was the adult survival assay conducted? In the M&M this is not mentioned
Why was only mortality of worms assessed? The worms can also show decreased viability, changing morphology etc
Results
Line 210: delete: This suggests that B. bifurcata has a dual anthelmintic activity, i.e., egg hatching and a direct nematicidal activity on freshly hatched larvae. This sentence does not belong into material and methods
I would suggest to harmonize the concentrations—all in ug/ml
A 6 days assay is very long, in light of a potential future clinical application this does not make sense.
I would suggest to determine also the IC50 of the extract (in 3.2 they are provided in 3.3)
3.3. geographical variation of algae: many parts of this section already discuss the results (should be moved to discussion)
Discussion:
Line 323: I think that the concentrations used are enormously high (and long incubation times) and the authors seem to pick examples of other studies working also with high concentrations. There are numerous examples in the literature where extracts of plants/algae showed an activity at lower concentrations. I think this is quite biased. For example the activity of Bifurcaria bifurcate against T. b. rhodesiense is much lower (https://pubmed.ncbi.nlm.nih.gov/20564461/)
Having said that I have the impression that the authors are trying to oversell their findings. The algae are also known to be cytotoxic (https://pubmed.ncbi.nlm.nih.gov/20564461/) and this was not evaluated in this study.
So, it would be good to tone down the findings a bit, place them in the right context and also discuss limitations
Conclusion: I do not think the next step would be in vivo studies, one should first try to find out which component(s) is responsible for the activity. A next step would be a bioassay guided fractionation. As the algea contains diterpenes with enormous structural diversity it would be interesting to elucidate more in this direction. Cytotoxicity studies are needed.
Author Response
"Please see the attachment."
